# Novel Gene Rearrangement and the Complete Mitochondrial Genome of *Cynoglossus monopus*: Insights into the Envolution of the Family Cynoglossidae (Pleuronectiformes)

**DOI:** 10.3390/ijms21186895

**Published:** 2020-09-20

**Authors:** Chen Wang, Hao Chen, Silin Tian, Cheng Yang, Xiao Chen

**Affiliations:** 1College of Marine Sciences, South China Agriculture University, Guangzhou 510642, China; wangchen2971@163.com (C.W.); tian991118@163.com (S.T.); aktt2365@163.com (C.Y.); 2Cell and Molecular Biology Program, University of Arkansas, Fayetteville, AR 72701, USA; chenhao910827@gmail.com; 3Guangdong Laboratory for Lingnan Modern Agriculture, South China Agriculture University, Guangzhou 510642, China

**Keywords:** *Cynoglossus monopus*, mitochondrial genome, novel rearrangement, intramitochondrial recombination, phylogenetic analyses, divergence times

## Abstract

*Cynoglossus monopus*, a small benthic fish, belongs to the Cynoglossidae, Pleuronectiformes. It was rarely studied due to its low abundance and cryptical lifestyle. In order to understand the mitochondrial genome and the phylogeny in Cynoglossidae, the complete mitogenome of *C. monopus* has been sequenced and analyzed for the first time. The total length is 16,425 bp, typically containing 37 genes with novel gene rearrangements. The tRNA-*Gln* gene is inverted from the light to the heavy strand and translocated from the downstream of tRNA-*Ile* gene to its upstream. The control region (CR) translocated downstream to the 3’-end of *ND1* gene adjoining to inverted to tRNA-*Gln* and left a 24 bp trace fragment in the original position. The phylogenetic trees were reconstructed by Bayesian inference (BI) and maximum likelihood (ML) methods based on the mitogenomic data of 32 tonguefish species and two outgroups. The results support the idea that Cynoglossidae is a monophyletic group and indicate that *C. monopus* has the closest phylogenetic relationship with *C. puncticeps*. By combining fossil records and mitogenome data, the time-calibrated evolutionary tree of families Cynoglossidae and Soleidae was firstly presented, and it was indicated that Cynoglossidae and Soleidae were differentiated from each other during Paleogene, and the evolutionary process of family Cynoglossidae covered the Quaternary, Neogene and Paleogene periods.

## 1. Introduction

Flatfishes (Pleuronectiformes) are unique animals with both eyes moved on one side of the body through asymmetrical development. Furtherly, according to the traditional morphological opinions, the tonguefishes (Cynoglossidae and Soleoidae) are the most specified and have an advanced classification in Pleuronectiformes [1,2]. *Cynoglossus monopus* belongs to Cynoglossidae in suborder Soleoidei, distributed from the Malay Archipelago to the Indian Ocean, including the west and northward along the South China Sea [1,2]. It is a small benthic fish with ctenoid scale covered on both-side, two lateral line on ocular-side and absent lateral line on blind-side. Especially, the small pedunculate eyes are the distinct diagnosis in *C. monopus* from other *Cynoglossus* species.

Traditionally, suborder Soleoidei includes three families: Achiridae, Cynoglossidae and Soleidae [3], among which Achiridae was thought to be separated from Soleidae and represented the primitive sister group to Soleidae and Cynoglossidae [4]. The Cynoglossidae was believed to be derived from the Soleidae as well, and have a closer relationship with Soleidae than Achiridae [5]. In morphology, both eyes of the species in Cynoglossidae and Soleidae turned to the left-side and the right-side, respectively, which is the principle basis to distinguish these two families [1]. Within Cynoglossidae, there are more than 140 species in three genera of two subfamilies. The genera *Cynoglossus* and *Paraplagusia* were sister groups belonging to subfamily Cynoglossinae, while *Symphurus* was the only genus in subfamily Symphurinae [3].

The mitochondrial DNA (mtDNA) is a double-stranded circular DNA molecule. Similar in composition and structure to most vertebrates, it is 15–20 kb in length, containing 13 protein-coding genes (PCGs), 2 ribosomal RNA genes (rRNA), 22 transfer RNA genes (tRNA), a light-strand replication origin (O_L_), and a control region (CR) that possesses cis-regulatory elements [6,7]. Compared to the nuclear genome, the mitochondrial genome shares maternal inheritance, large copy number, stable gene composition, conserved gene arrangement, and high evolutionary rate [6,8]. Therefore, mitogenome has been widely used to infer phylogenetic relationships and population genetics in animal [9,10,11]. Furthermore, the gene rearrangements have been reported in various vertebrates [11,12,13,14,15,16], but the organization in most fish mitogenomes is generally considered quite conserved [17,18]. However, we analyzed the complete mitogenome sequence of *C. monopus* and found a novel gene order in this study. Although there are some phylogenetic studies focusing on genus *Cynoglossus*, the evolutionary histories of Cynoglossidae and Soleidae are still remain unclear since very few reports concentrated on the divergence time among them (Table 1). To fill the gap in genetic information, this study analyzes the phylogeny and evolutionary histories of Cynoglossidae and Soleidae based on the complete mitogenome of *C. monopus* and all available mitogenomic data of these two families.

## 2. Results and Discussions

### 2.1. Genome Organization and Nucleotide Composition

The complete mitochondrial genome of *C. monopus* is 16,425 bp in length (GenBank accession number MT798589), within the range of other reported Pleuronectiformes mitogenomes from 15,973 bp (*Kareius bicoloratus*) to 18,369 bp (*Cynoglossus trigrammus*). This mitogenome contains 13 protein-coding genes, 22 tRNA genes, 2 rRNA genes, the origin of light-strand replication (O_L_) and a control region (CR) that possesses cis-regulatory elements (Figure 1, Table 2). Except *ND6* and eight tRNA genes encoded on the Light-strand, others are encoded on the Heavy-strand. The tRNA-*Gln* gene was inverted from the L-strand position to the H-strand in the other position. This result is consistent with the findings of previous studies where species in subfamily Cynoglossinae have been found to have large-scale gene rearrangements, and a unique gene order, CR-*Gln*-*Ile*-*Met*, which is different from the typical gene order of CR-*Phe*-12S-*Val*-16S-*Leu1*-*ND1*-*Gln-Ile-Met* [19,20,21,22,23,24,25,26,27].

The overall base composition is 30.80% A, 24.04% C, 14.77% G, and 30.39% T, with a high AT content (61.19%). The AT-skew and GC-skew of the *C. monopus* mitogenome are 0.01 and −0.24, respectively (Table 3). *C. monopus* showed a higher AT content (63.19%) in 16S rRNA gene than the 12S rRNA (55.31%). The control region is 736 bp in length with the rich AT (70.92%) and poor G (11.41%) content. Moreover, the AT content of 13 PCGs ranged from 54.21% (*ND4L*) to 66.67% (*ATP8*). The AT content of tRNAs is 61.10% in average, while the CG content is 38.90%.

### 2.2. Protein-Coding Genes (PCGs), Transfer RNAs and Ribosomal RNAs

In the mitogenome of *C. monopus*, except for the *ND3,* started with ATT and *COII* terminated with T, most PCGs have typical initiation codons (ATG or GTG) and termination codons (TAA or TAG) (Table 2). The size of 13 PCGs ranged from 165 bp (*ATP8*) to 1854 bp (*ND5*), and the total length of PCGs is 11,422 bp, similar to other Cynoglossidae species (Figure 2B). The GC-skews of the 13 PCGs were all negative and the majority of the AT-skew values were negative, similarly. The AT skews of *ND3* and *ND6* were the lowest and the GC-skews of *COI* and *COII* were the highest (Table 3).

The 22 tRNA genes of *C. monopus* ranged from 65 bp (tRNA-*Cys*) to 76 bp (tRNA-*Lys*), with a total of 1540 bp. The 12S rRNA (946 bp) and 16S rRNA genes (1702 bp) of the *C. monopus* mitogenome are located in the typical position between tRNA-*Phe* and tRNA-*Leu* (UUR), and separated by tRNA-*Val*, with a high AT content of 60.42%. Additionally, the origin of light-strand is 24 bp in length between tRNA-*Asn* and tRNA-*Cys*.

### 2.3. Mitochondrial Gene Codon Usage and Skewness

The amino acids were utilized by either two or four different codons, respectively. Ile was the most frequently used, while *Met* and *Trp* were the least frequently used (Figure 3A). In addition, the relative synonymous codon usage (RSCU) analysis indicated that *Arg* (AGG, AGA), *Pro* (CCC) and *Ala* (GCU) were the most frequent, and *Ser* (UCG), *Thr* (ACG), *Pro* (CCG) and *Lys* (AAG) were rare.

Moreover, the AT-skew and GC-skew in the mitogenome of the subfamily Cynoglossinae were reflected in the codon usage consistently (Figure 3B). The RSCU values indicated that codons with A or U in the third position were more frequent than C or G. Additionally, the GC-skew values are negative in the available mitogenomes of Cynoglossidae species (Figure 2A). In contrast, most of the AT skews are positive, except for *C. itinus*.

### 2.4. Gene Rearrangement

Similar to other *Cynoglossus* and *Paraplagusia* species which mitogenome available, the CR of *C. monopus* is translocated downstream to the place between *ND1* and tRNA-*Gln* instead of the typical location between tRNA-*Pro* and tRNA-*Phe* (Figure 1). The translocation left a functionally undefined 24 bp trace fragment in the original CR position. Furthermore, the tRNA-*Gln* gene (Q) translocated from the downstream of tRNA-*Ile* gene (I) to its upstream with inverted encoding direction (Q’). The result was the formation of the Q’-I-M gene order, which is different from the typical I-Q-M gene order in most vertebrates (Figure 4a).

Among several mitochondrial gene rearrangement models [46,47,48], the intramitochondrial recombination is the most probable mechanism to explain the rearrangement events in the mitogenome of *C. monopus* based on the principle of parsimony. The hypothesized intermediate steps are as follows. Firstly, the whole Control region (likely carrying some neighbour sequences) translocated to the downstream of *ND1* gene, and left a duplicated partial Control region in the original position. At the same time, the tRNA-*Gln* gene was inversely translocated to the upstream of tRNA-*Ile* with some neighbour sequences and left a duplicated partial tRNA-*Gln* fragment between tRNA-*Ile* and tRNA-*Met*. The rearrangement formed a new *ND1*-CR-Q’-I-M region in the mitogenome of *C. monopus* with unfunctional gene fragments connecting each gene (Figure 4b). Secondly, after a rapid deletion process under the strong selective pressure, the unfunctional gene fragments in each gene junction left a 24 bp trace fragment in the original CR position and 5 and 6 bp intergenic spacers between Q’-I and I-M gene junctions, respectively (Figure 4c,d).

### 2.5. Phylogenetic Analyses

Both BI and ML analyses produced almost identical topologies with similar branch lengths. Most clades were strong supported by high bootstraps (ML) and posterior probabilities (Bayesian) values (Figure 5). Contrary to the traditional classification [36,39,40], the molecular phylogenetic tree showed a clade with three Soleidae species (*P. pavoninus*, *A. kobensis* and *L. melanospilos*) clustered to Cynoglussidae as a sister group rather than other Soleidae species (Figure 5). This suggests that Soleidae is not a monophyletic group. However, the insufficient species could cause questionable phylogeneic results [49], and as Soleidae is a large family including 32 genera and at least 180 species [50], more species will be necessarily needed for further investigation. Furthermore, the most terminal branch relationship within *Cynoglossus* is consistent with traditioanal taxon. This supports the opinion that lateral line number and scale characteristics are valuable morphological diagnoses in classification [1].

Among Cynoglussidae, compared to other *Cynoglossus* species, *C. monopus* had the closest phylogenetic relationship to *C. puncticeps*. Traditionally, they were also classified into the subgenus Cynoglossoides by sharing the most similarities with both sides covered with ctenoid scale, two lateral lines on the ocular-side and an absent lateral line on the blind-side [1]. However, the clade of *C. monopus* and *C. puncticeps* clustered to the *Paraplagusia* rather than other *Cynoglossus* species. This suggests that the relationship between *Cynoglossus* and *Paraplagusia* is more complex than expected and needs further study.

### 2.6. Divergence Time Analyses

The advent of the phylogenomic era has significantly improved our understanding of the taxonomy and phylogenetic relationships of many animals [51]. Compared with other flatfish studies, the divergence time estimations within or between families Cynoglossidae and Soleidae were additionally conducted based on two calibration constraints in this study. The divergence time between Cynoglossidae and Soleidae could be occurred at about 45.15 Mya (37.47–49.99 Mya). Furthermore, the divergence time of the subfamily Cynoglossinae could be dated back to 20.92 Mya (17.11–28.44 Mya) earlier than the fossil record of *C. leuchsi* (lower and middle Miocene) [5].

The Chronogram for the 34 species of Pleuronectiformes covered three geological epochs, including the Quaternary, Neogene, and Paleogene periods (Figure 6). It was indicated that family Soleidae (33.90 Mya (14.79–49.30 Mya)) evolved earlier than family Cynoglossidae (22.82 Mya (17.11–28.44 Mya)). The fossil-calibrated divergence time estimated that genus *Paraplagusia* diverged at 22.82 Mya (17.11–28.44 Mya) within the Neogene as the most primitive species in family Cynoglossidae. During this period, *C. monopus* and *C. puncticeps* began to differentiate from other species at about 14.29 Mya (6.43–22.04 Mya). In particular, we found that most species of family Soleidae were divided during the Neogene period and the speciation of family Cynoglossidae was happening during the Quaternary period. More importantly, large-scale gene rearrangement was recently detected to occur in genera *Cynoglossus* and *Paraplagusia*, and they were differentiated from the clade of Soleidae (*P. pavoninus*, *A. kobensis* and *L. melanospilos* as a subgroup) about 40.53 Mya (31.87–49.99 Mya). However, another clade of family Soleidae found no evidence of gene rearrangement, and the two clades began to differentiate from each other at about 45.15 Mya (37.47–49.99 Mya).

## 3. Materials and Methods

### 3.1. Specimen Collection and DNA Extraction

A single specimen of *C. monopus* was collected from Sanya (E 108°56′, N 18°09′), Hainan Province. The voucher specimen (Voucher No. HNSY2010060432) was deposited in the College of Marine Sciences, South China Agricultural University, Guangzhou, China (SCAU). Animal experiments were conducted in accordance with the guidelines and approval of the Animal Research and Ethics Committees of SCAU. Genomic DNA was extracted from the muscle of *C. monopus* according to the standard phenol-chloroform procedure [52]. The data analysis method is based on the previous study [53].

### 3.2. PCR Amplification and Sequencing

To ensure a sufficient amount of DNA for the amplification and sequencing of these small fishes, the parameters of the LA-PCR reactions were mostly in accordance with the manufacturer’s recommendations. PCR products were purified using the gel purification kit (Invitrogen) after gel-cutting (1.5% TBE agarose). Purified PCR products were sequenced directly on an ABI 3730 automated sequencer (Life Technologies Holdings Pte Ltd, Tuas, Singapore) with ABI PRISM BigDye Terminators v3.0 Cycle Sequencing (ABI) using the primer-walking strategy. The eight fragments were separated from the complete mitogenome of *C. monopus*, with universal primer for Cynoglossidae mitogenome.

### 3.3. Sequene Analysis

Sequence data were analyzed and compiled to create the complete genome using the SeqMan program from Lasergene soft package (DNASTAR, Madison, WI, USA), and manually adjusted in a few cases. The complete mitogenome was annotated using the software of Sequin v16.0 (National Library of Medicine, Bethesda, MD, USA). Mitochondrial tRNA genes and their secondary structures were obtained by ARWEN v1.2 [54], and identified by tRNAscan-SE Search Server v2.0 (Washington University School of Medicine, St Louis, MO, USA) [55] using default search mode, then anticodons were further confirmed. Annotation and accurate boundary determination of protein-coding and ribosomal RNA genes were first performed by NCBI-BLAST searches, and then by alignment and manual comparisons with the other released reference mitogenomes of Cynoglossidae species using DNAMAN v6.0 (Lynnon Biosoft, San Ramon, CA, USA).

The complete mitogenome of *C. monopus* was uploaded to GenBank with accession number MT798589. The graphical genome map of the *C. monopus* mitogenome was drawn using CGView Server v1.0 [56]. The base composition, codon usage and RSCU values were obtained using MEGA 7.0 (Tokyo Metropolitan University, Tokyo, Japan) [57]. Strand asymmetry was estimated using the following formulas by Perna and Kocher (1995) [58]: AT skew = [A − T]/[A + T] and GC skew = [G − C]/[G + C].

### 3.4. Phylogenetic and Divergence Time Analyses

Phylogenetic analyses were conducted using 19 Cynoglossidae species and 13 Soleidae species with *P. erumei* and *P. olivaceus* as outgroups. All sequences were available in GenBank (27/7/2020). We aligned DNA sequences of 12 protein-coding genes (except *ND6*) and two rRNA genes in the 34 species using the MAFFT program with the default parameters [59]. The alignments of PCGs (except *ND6*) excluded the start codon and the stop codon. Then, ambiguously aligned fragments of two alignments were removed using Gblocks [60], and exported two gblocks alignments. We used a dataset comprised of concatenated the gblocks of 12 protein-coding genes (the first and second codon positions, except *ND6*) and 2 rRNA genes.

Then, the data were divided into three pre-defined partitions for the best partitioning scheme using PartitionFinder 2.0 (Macquarie University Genes to Geosciences Centre, North Ryde, NSW, Australia) [61], with the greedy algorithm. ModelFinder [62] lugin integrated into PhyloSuite v1.2.1 (Bio-Transduction Lab, Wuhan, China) [63] was used to select the best-fit partition model (Edge-unlinked). The best-fit model according to BIC: GTR + F + G4 was selected as the optimal model for the glocks of first codons of PCGs (except *ND6*) and rRNAs, respectively, whereas TVM + F + G4 was chosen for the gblocks of two rRNA genes. Additionally, phylogenetic analyses were performed using the Bayesian analyses (BI) and Maximum Likelihood (ML) methods [64,65]. ML analysis was inferred using IQ-TREE v1.6.2 [66] plugin integrated into PhyloSuite v1.2.1 under Edge-linked partition model for 10,000 ultrafast [67] bootstraps, approximate Bayes test [64], as well as the Shimodaira–Hasegawa-like approximate likelihood-ratio test [68]. Additionally, Bayesian inference with partition model was conducted in MrBayes 3.2.6 (University of California, La Jolla, San Diego, CA, USA) [65] under the partition model (two parallel runs, 2,000,001 generations), in which the initial 25% of sampled data were discarded as burn-in with default settings and 5 × 106 metropolis-coupled Markov chain Monte Carlo (MCMC) generations. The iTOL dataset files produced by PhyloSuite were then used to visualize and annotate the phylograms and gene orders in iTOL [69].

The evolutionary analysis was inferred by BEAST v1.10.4 (open source under the GNU lesser general public license) using Bayesian Information Criterion based on the two gblocks (the first and second codon positions of PCGs, except *ND6*) and two rRNA genes of 34 Pleuronectiformes species [70]. The divergence times were presented in the Time Tree database (http://www.timetree.org/) [71] and fossil-based comparative analyses [72]. The time tree was computed using two calibration constraints: the most recent common ancestor (MRCA) of genus *Cynoglossus* was estimated to be lower and middle Miocene period based on fossilized *C. leuchsi* and the MRCA of both Cynoglossidae and Soleidae was estimated to be at the lower Eocene at least 45 Mya [5]. Molecular dating involved a Birth-Death process as the tree prior, and an uncorrelated relaxed clock as the best model. The chains of 1 × 10^8^ samples were run for the MCMC analysis, and the 10% of all samples was burn-in using TreeAnnotator. Tracer v1.7.1 was used to confirm the output [73]. FigTree v1.4.3 was used to edit the results.

## 4. Conclusions

This study suggests that mitochondrial gene rearrangement in Cynoglossidae is a single originated evolutionary event, which occurred in the common ancestor of *Cynoglossus* and *Paraplagusia* before at least 22.82 Mya (17.11–28.44 Mya). The highly similar rearranged gene order in all available *Cynoglossus* and *Paraplagusia* species [18,35,36,37,38,39,40,41,42,43,44,45] inferred that novel gene order possesses some selective advantage in this group, then kept it conserved in the whole lineage. Even the details of the molecular mechanism are still unclear; the intramitochondrial recombination is the most probable model to explain the process of gene rearrangement in this group based on the principle of parsimony.

The phylogenetic relationships constructed by ML and BI method are consistent. The time tree covers three geological epochs, including the Quaternary, Neogene, and Paleogene periods. The divergence times show that *C. monopus* and *C. puncticeps* begin to differentiate from other species at about 14.29 Mya (6.43–22.04 Mya) within the Neogene period. Tonguefish is a highly specialized body. The traditional morphological classifications of tonguefish are often controversial due to the unstable diagnosis caused by its asymmetrical development. The mitochondrial rearrangement is a helpful gene marker to reconstruct the phylogenetic relationship in this group with mitogenome and other molecular data.

## Figures and Tables

**Figure 1 ijms-21-06895-f001:**
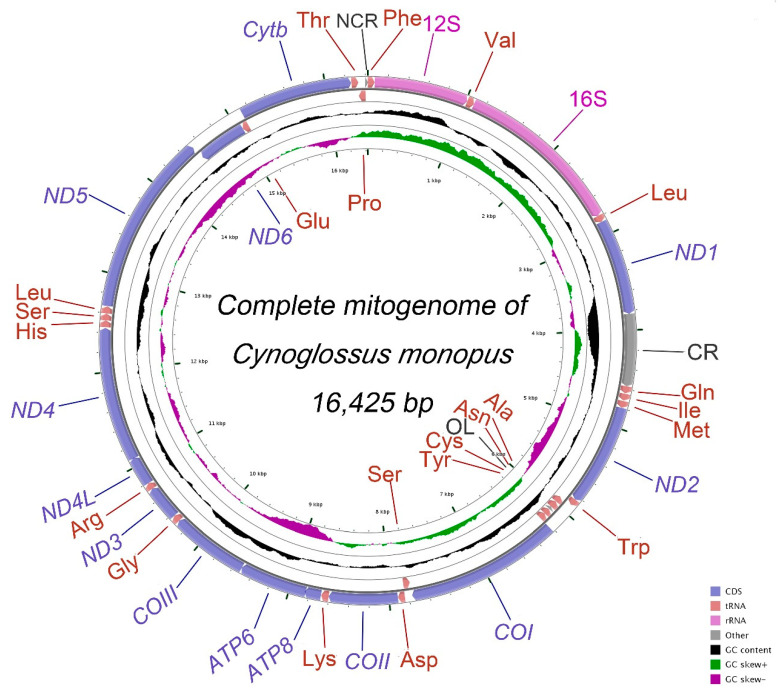
Graphical genome map of the mitogenome of *C. monopus.* The genes outside the outermost circle are transcribed clockwise, whereas the genes inside the outermost circle are transcribed counterclockwise. The inside circle shows the GC content and GC skews.

**Figure 2 ijms-21-06895-f002:**
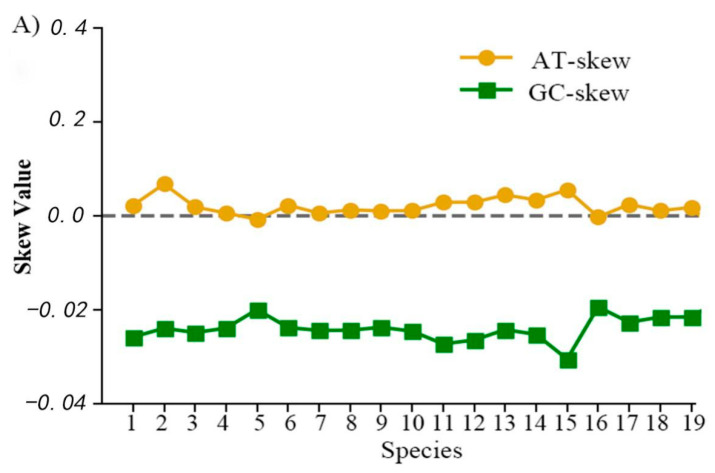
Graphical illustration shown the usage bias of AT and GC (AT-skew and GC-skew values) (**A**), and the length of protein-coding genes (PCGs), tRNAs, rRNAs, and control regions (**B**) in the mitogenomes of 19 species in the subfamily Cynoglossinae. Note: Species 1–21: *C. abbreviatus*, *C. bilineatus*, *C. gracilis*, *C. interruptus*, *C. itinus*, *C. joyneri*, *C. monopus*, *C. nanhaiensis*, *C. puncticeps*, *C. robustus*, *C. roulei*, *C. semilaevis*, *C. senegalensis*, *C. sinicus*, *C. trigrammus*, *C. zanzibarensis*, *P. bilineata*, *P. blochii* and *P. japonica.*

**Figure 3 ijms-21-06895-f003:**
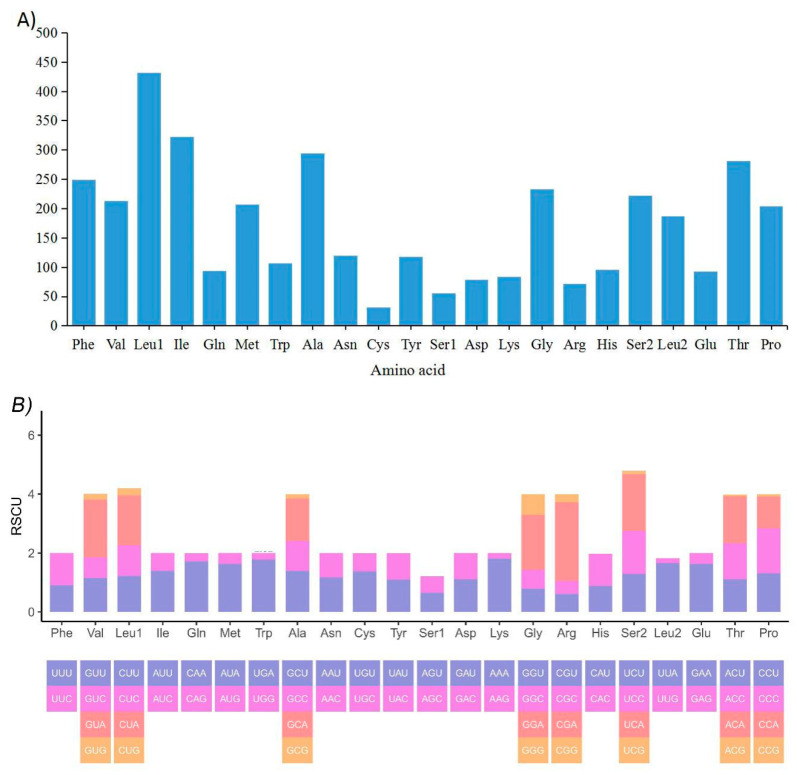
Amino acid composition (**A**) and relative synonymous codon usage (**B**) in the mitogenome of *C. monopus*.

**Figure 4 ijms-21-06895-f004:**
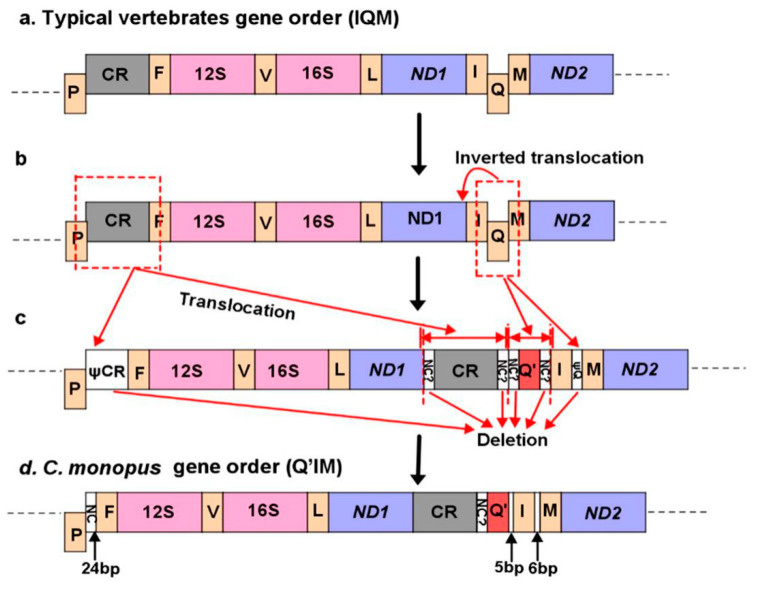
Inferred gene rearrangement between the gene order of typical vertebrates and the mitogenome of *C. monopus*. The typical vertebrate gene order (**a**); inferred intermediate processes of gene rearrangement (**b**,**c**); the gene order in the mitogenome of *C. monopus* (**d**).

**Figure 5 ijms-21-06895-f005:**
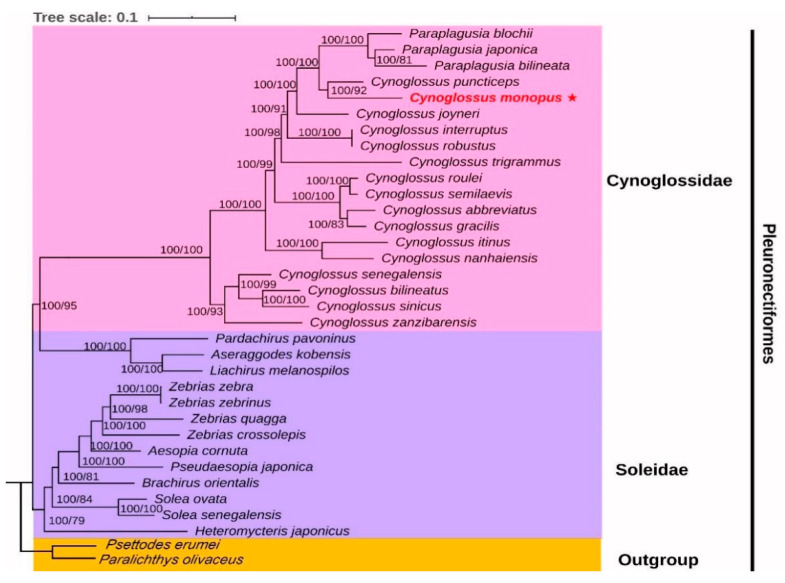
Phylogenetic tree of *C. monopus* was performed using partial genomes of 32 species of Pleuronectiformes and *Psettodes erumei* and *Paralichthys olivaceus* were used as outgroups, with Bayesian analyses and Maximum likelihood analyses. Species in red indicates sequence generated in this study. Bootstrap support (**right**) and Bayesian posterior probability values (**left**) of each clade are displayed next to the nodes.

**Figure 6 ijms-21-06895-f006:**
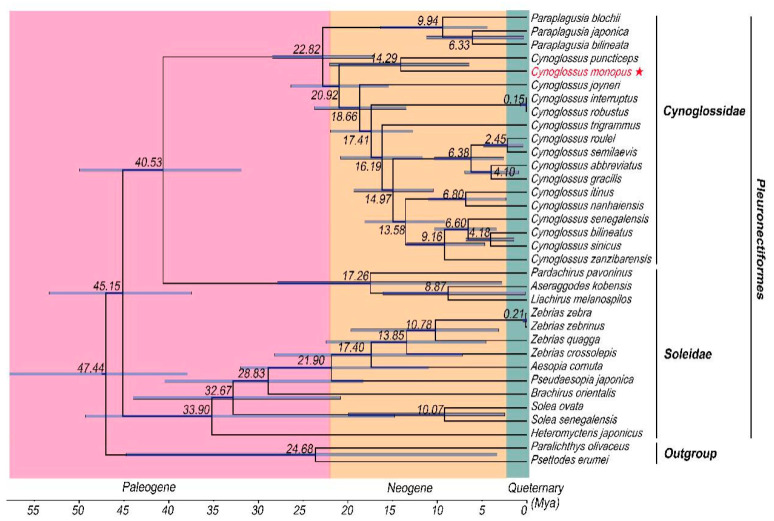
Chronogram for the 32 species of Pleuronectiformes with *P. erumei* and *P. olivaceus* as outgroups based on the concatenated nucleotide sequences of the 12 PCGs (except *ND6* and the third codon positions) and 2 rRNA genes using BEAST analysis. Numbers near the nodes indicate the average estimated divergence time estimated in Mya, and the 95% confidence intervals for each node are shown in blue bars.

**Table 1 ijms-21-06895-t001:** Details of species and mitogenomes of Pleuronectiformes used in this study.

Family	Genus	Species	Length (bp)	Accession ID	Reference
Cynoglossidae	Cynoglossus	*Cynoglossus abbreviatus*	16,417	JQ349004	[23]
		*Cynoglossus bilineatus*	16,454	JQ349000	[24]
		*Cynoglossus gracilis*	16,565	KT809367	[25]
		*Cynoglossus interruptus*	17,262	LC482306	[20]
		*Cynoglossus itinus*	16,915	JQ639062	Unpublished
		*Cynoglossus joyneri*	16,428	NC030256	[27]
		*Cynoglossus monopus*	16,425	MT798589	This study
		*Cynoglossus nanhaiensis*	17,130	MT117229	[28]
		*Cynoglossus puncticeps*	17,142	JQ349003	[26]
		*Cynoglossus robustus*	16,720	LC482305	[21]
		*Cynoglossus roulei*	16,565	MN966658	[22]
		*Cynoglossus semilaevis*	16,731	EU366230	[19]
		*Cynoglossus senegalensis*	16,519	MH709122	[29]
		*Cynoglossus sinicus*	16,478	JQ348998	[30]
		*Cynoglossus trigrammus*	18,369	KP057581	[31]
		*Cynoglossus zanzibarensis*	16,569	KJ433559	[32]
	Paraplagusia	*Paraplagusia bilineata*	16,985	NC023227	Unpublished
		*Paraplagusia blochii*	16,611	JQ349002	[33]
		*Paraplagusia japonica*	16,694	JQ639066	[34]
Soleidae	Aesopia	*Aesopia cornuta*	16,737	KF000065	[35]
	Aseraggodes	*Aseraggodes kobensis*	16,944	KJ601760	[36]
	Brachirus	*Brachirus orientalis*	16,600	KJ513134	[37]
	Heteromycteris	*Heteromycteris japonicus*	17,111	JQ639060	[38]
	Liachirus	*Liachirus melanospilos*	17,001	KF573188	[39]
	Pardachirus	*Pardachirus pavoninus*	16,573	KJ461620	[40]
	Pseudaesopia	*Pseudaesopia japonica*	16,789	KJ433482	[41]
	Solea	*Solea ovata*	16,782	KF142459	[42]
		*Solea senegalensis*	16,659	AB270760	[18]
	Zebrias	*Zebrias crossolepis*	16,734	KJ433564	[43]
		*Zebrias quagga*	17,045	NC023225	[44]
		*Zebrias zebra*	16,758	JQ700100	[45]
		*Zebrias zebrinus*	16,762	KC491209	Unpublished
Paralichthyidae	Paralichthys	*Paralichthys olivaceus*	17,090	NC002386	Unpublished
Psettodidae	Psettodes	*Psettodes erumei*	17,315	FJ606835	Unpublished

**Table 2 ijms-21-06895-t002:** Features of the *C. monopus* mitochondrial genome.

Gene	Strand	Position		Size (bp)	Amino Acids (aa)	Codon		Anti-Codon	Intergenic Nucleotides (bp)
		From	To			Start	Stop		
tRNA-*Phe*	H	1	68	68				GAA	0
12S rRNA	H	69	1014	946					0
tRNA-*Val*	H	1015	1087	73				TAC	0
16S rRNA	H	1088	2789	1702					0
tRNA-*Leu*	H	2790	2859	70				TAA	0
*ND1*	H	2860	3834	975	324	ATG	TAA		0
Control region	H	3835	4570	736					0
tRNA-*Gln*	H	4571	4643	73				GAT	0
tRNA-*Ile*	H	4649	4717	69				TTG	5
tRNA-*Met*	H	4724	4793	70				CAT	6
*ND2*	H	4795	5838	1044	347	ATG	TAA		1
tRNA-*Trp*	H	5838	5905	68				TCA	−1
tRNA-*Ala*	L	5908	5976	69				TGC	2
tRNA-*Asn*	L	5979	6051	73				GTT	2
O_L_	L	6056	6079	24					4
tRNA-*Cys*	L	6086	6150	65				GCA	6
tRNA-*Tyr*	L	6151	6218	68				GTA	0
*COI*	H	6220	7770	1551	516	GTG	TAA		1
tRNA-*Ser*	L	7771	7841	71				TGA	0
tRNA-*Asp*	H	7844	7912	69				GTC	2
*COII*	H	7914	8604	691	230	ATG	T		1
tRNA-*Lys*	H	8605	8680	76				TTT	0
*ATP8*	H	8683	8847	165	54	ATG	TAA		2
*ATP6*	H	8838	9521	684	227	ATG	TAA		−10
*COIII*	H	9521	10,306	786	261	ATG	TAA		−1
tRNA-*Gly*	H	10,306	10,375	70				TCC	−1
*ND3*	H	10,376	10,726	351	116	ATG	TAA		0
tRNA-*Arg*	H	10,725	10,793	69				TCG	−2
*ND4L*	H	10,794	11,090	297	98	ATG	TAA		0
*ND4*	H	11,084	12,448	1365	454	ATG	TAA		−7
tRNA-*His*	H	12,456	12,524	69				GTG	9
tRNA-*Ser*	H	12,525	12,592	68				GCT	0
tRNA-*Leu*	H	12,595	12,667	73				TAG	2
*ND5*	H	12,671	14,524	1854	617	ATG	TAA		3
*ND6*	L	14,530	15,051	522	183	ATG	TAG		5
tRNA-*Glu*	L	15,052	15,120	69				TTC	0
*Cytb*	H	15,123	16,259	1137	378	ATG	TAG		2
tRNA-*Thr*	H	16,263	16,331	69				TGT	3
tRNA-*Pro*	L	16,331	16,401	71				TGG	−1
Noncoding region	H	16,402	16,425	24					0

**Table 3 ijms-21-06895-t003:** Nucleotide composition and skewness levels calculated for sequenced majority strand of the *C. monopus*.

Regions	Size (bp)	Nucleotides Composition (%)						AT-Skew	GC-Skew
		T	C	A	G	AT	GC		
Whole genome	16,425	30.39	24.04	30.80	14.77	61.19	38.81	0.01	−0.24
PCGs	11,422	32.38	25.03	28.38	14.21	60.76	39.24	−0.07	−0.28
1st codon position	3808	24.91	24.52	27.10	23.46	52.02	47.98	0.04	−0.02
2nd codon position	3807	42.10	26.48	18.15	13.27	60.25	39.75	−0.40	−0.33
3rd codon position	3807	30.94	24.06	38.92	6.08	69.86	30.14	0.11	−0.60
*ATP6*	684	30.85	31.43	27.49	10.23	58.33	41.67	−0.06	−0.51
*ATP8*	165	32.73	25.45	33.94	7.88	66.67	33.33	0.02	−0.53
*COI*	1551	31.72	23.86	26.69	17.73	58.41	41.59	−0.09	−0.15
*COII*	691	32.27	22.43	30.39	14.91	62.66	37.34	−0.03	−0.20
*COIII*	786	30.53	27.35	25.83	16.28	56.36	43.64	−0.08	−0.25
*Cytb*	1137	34.21	25.77	26.30	13.72	60.51	39.49	−0.13	−0.31
*ND1*	975	32.51	24.62	28.41	14.46	60.92	39.08	−0.07	−0.26
*ND2*	1044	31.90	26.44	31.32	10.34	63.22	36.78	−0.01	−0.44
*ND3*	351	32.76	26.21	27.35	13.68	60.11	39.89	−0.09	−0.31
*ND4*	1365	32.60	24.62	30.26	12.53	62.86	37.14	−0.04	−0.33
*ND4L*	297	29.97	29.29	24.24	16.50	54.21	45.79	−0.11	−0.28
*ND5*	1854	31.01	25.57	30.91	12.51	61.92	38.08	0.00	−0.34
*ND6*	522	41.38	12.26	21.65	24.71	63.03	36.97	−0.31	0.34
tRNAs	1540	30.19	17.92	30.91	20.97	61.10	38.90	0.01	0.08
rRNAs	2648	25.11	20.85	35.31	18.73	60.42	39.58	0.17	−0.05
Control region	736	35.87	17.66	35.05	11.41	70.92	29.08	−0.01	−0.21

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
