# Peer review of "Novel Gene Rearrangement and the Complete Mitochondrial Genome of Cynoglossus monopus: Insights into the Envolution of the Family Cynoglossidae (Pleuronectiformes)"

_ijms, 2020, doi:10.3390/ijms21186895_

Round 1

Reviewer 1 Report

The description of a novel gene order in mtDNA, the discussion about it putative origin, and the phylogeny presented are really interesting and well described. The paper is interesting, but some points are not clear.

My three points:

First of all I don’t understand why the authors use the expression ‘species complex’ if we just have one species without evidence of cryptic species in the same name.

In the line 42/43 the authors say: “Recently, we released a high coverage genome sequence for Cynoglossus monopus and found a novel gene order, similar to other Pleuronectiformes [18].”- I understand that this sequence was only released for GenBank purposes and not already available because if the information is already public it doesn’t make sense to publish it again in the present paper.

The phylogenetic analyses seems ok, but the determination of divergence time was  not properly analyzed/described. The use of MEGA to determinate the divergence time is not usual because it’s a very limited program in this point. There’s many other very good software that could be used. But the main issue is how the was calibrated. The authors need to better explain this and use fossil or geological information to do these analyses.

Author Response

Dear Editors and Reviewers:

Thank you for the reviewers’ comments concerning our manuscript entitled “Phylogenetic Relationship and Gene Rearrangement of the Complete Mitochondrial Genome of Cynoglossus monopus (Pleuronectiformes: Cynoglossidae), with a Comparative Analysis of Other Flatfishes” (ID: ijms-914162). Those comments are all valuable and very helpful for revising and improving our paper, as well as the important guiding significance to our researches. We will re-structure the paper to improve clarity. We will also add more details and tried our best to improve the manuscript and made many changes in the manuscript, which we hope meet with approval. Revised portion are marked in red in the paper. The main corrections in the paper and the responds to the reviewer’s comments are as flowing:

Point 1: First of all, I don’t understand why the authors use the expression ‘species complex’ if we just have one species without evidence of cryptic species in the same name. 

Response 1: The expression is a concern. We have re-written this part according to the Reviewer’s suggestion.

Point 2: In the line 42/43 the authors say: “Recently, we released a high coverage genome sequence for Cynoglossus monopus and found a novel gene order, similar to other Pleuronectiformes [18].”- I understand that this sequence was only released for GenBank purposes and not already available because if the information is already public it doesn’t make sense to publish it again in the present paper.

Response 2: We have made correction according to the Reviewer’s comments. And this sequence is to be held confidential until: Jun 30, 2020. It doesn’t be released or published to public database until the data or accession number appears in print.

Point 3: The phylogenetic analyses seems ok, but the determination of divergence time was not properly analyzed/described. The use of MEGA to determinate the divergence time is not usual because it’s a very limited program in this point. There’s many other very good software that could be used. But the main issue is how the was calibrated. The authors need to better explain this and use fossil or geological information to do these analyses.

Response 3: Considering the Reviewer’s suggestion, we have chosen BEAST v1.10.4 to determinate the divergence time based on two fossil calibrations and have re-written this part. And delete some species to calibrate the fossil. So we also have re-written phylogenetic analyses and adjusted to relevanted parts of the article.

Once again, thank you very much for your comments and suggestions.

Reviewer 2 Report

Date: August 2020, 2020

Journal: International Journal of Molecular Sciences
Manuscript ID: ijms-914162
Type of manuscript: Article
Title: Phylogenetic Relationship and Gene Rearrangement of the Complete Mitochondrial Genome of Cynoglossus monopus (Pleuronectiformes: Cynoglossidae), with a Comparative Analysis of Other Flatfishes
Authors: Chen Wang, Hao Chen, Silin Tian, Cheng Yang, Xiao Chen * Submitted
to section: Molecular Genetics and Genomics,

Reviewer's report

The authors have sequenced complete mitochondrial genome sequence of the tongue sole Cynoglossus monopus. Combined with mitochondrial genomes from Genbank, they used the Cynoglossus monopus genome to reconstruct phylogenetic relationships of Cynoglossidae and Soleidae species.

There is a big problem with this manuscript concerning the quality of English language. It is not acceptable for the journal like IJMS.

On the whole, I think the paper could merit publication in IJMS, but after major revision.

Main points:

The mitochondrial genome annotation and sequence analysis are fine, but the divergence time analysis is not convincing enough. The authors used MEGA7.0 to reconstruct the evolutionary history of Pleuronectiformes. The chronogram (Figure 6) has no confidential intervals. The program BEAST2 (http://www.beast2.org/) is more appropriate for this purpose. At least, it is important to compare the results using the two programs.

Lines 138-149: the model of the gene rearrangements in the mitogenome of C. monopus is fully speculative. If the authors wish to discuss this question, they need to obtain clear arguments supporting the model.

Lines 191-199: the statement concerning “rapid evolution” is fully speculative. If the authors wish to discuss this question, they need to obtain clear arguments supporting the model.

Line 214: small parasites or fishes?

Lines 242-254: the text is highlighted. Revise.

Figures: the resolution of figures is low and should be enhanced. Figure legends should contain all details presented on figure.

Author Response

Dear Editors and Reviewers:
Thank you for the reviewers’ comments concerning our manuscript entitled “Phylogenetic Relationship and Gene Rearrangement of the Complete Mitochondrial Genome of Cynoglossus monopus (Pleuronectiformes: Cynoglossidae), with a Comparative Analysis of Other Flatfishes” (ID: ijms-914162). Those comments are all valuable and very helpful for revising and improving our paper, as well as the important guiding significance to our researches. We thank for your advice, we have done substantial revision and massive modifications of our paper,including each part of our paper and the language. we have done substantial revision and massive modifications of our paper including each part of our paper and the language. Revised portion are marked in red in the paper. The main corrections in the paper and the responds to the reviewer’s comments are as flowing:

Point 1: The mitochondrial genome annotation and sequence analysis are fine, but the divergence time analysis is not convincing enough. The authors used MEGA7.0 to reconstruct the evolutionary history of Pleuronectiformes. The chronogram (Figure 6) has no confidential intervals. The program BEAST2 (http://www.beast2.org/) is more appropriate for this purpose. At least, it is important to compare the results using the two programs.

Response 1: Considering the Reviewer’s suggestion, we have chosen BEAST v1.10.4 to determinate the divergence time based on two fossil calibrations and re-written this part. And delete some species to calibrate the fossil. So we also have re-written phylogenetic analyses and adjusted to relevanted parts of the article. (Lines 157-211)

Point 2: Lines 138-149: the model of the gene rearrangements in the mitogenome of C. monopus is fully speculative. If the authors wish to discuss this question, they need to obtain clear arguments supporting the model.

Response 2: We have re-written this part according to the Reviewer’s comments. And we choose the intramitochondrial recombination model to explain the gene rearrangements in the mitogenome of C. monopus. (Line 132-152)

Point 3: Lines 191-199: the statement concerning “rapid evolution” is fully speculative. If the authors wish to discuss this question, they need to obtain clear arguments supporting the model.

Response 3: Considering the Reviewer’s suggestion, we have re-written this part. (Line132-153)

Point 4: Line 214: small parasites or fishes?

Response 4: We are very sorry for our incorrect writing and have revised fishes.  (Lines 222)

Point 5: Lines 242-254: the text is highlighted. Revise.

Response 5: We are very sorry for our negligence of the highlighted part. We have revised. (Line 242-254)

Point 6: Figures: the resolution of figures is low and should be enhanced. Figure legends should contain all details presented on figure.

Response 6: As Reviewer suggested that we have enhanced the resolution of figures, and revised Figure legends.

We apologize for not clarifying all questions given the limited space and many reviews. We will thoroughly check and fix grammatical errors in the final submission. Once again, thank you very much for your comments and suggestions.

Round 2

Reviewer 1 Report

The text is interesting but need some improvements. I'm sending the text with some questions and marks. 

Author Response

Dear Reviewers:

Thank you for your comments concerning our manuscript. Those comments are all valuable and very helpful for revising and improving our paper, as well as the important guiding significance to our researches. We have studied comments carefully and have made correction which we hope meet with approval. Revised portion are marked in red in the paper. The main corrections in the paper and the responds to the reviewer’s comments are as flowing:

Point 1: The title. 

Response 1: Considering the Reviewer’s suggestion, we have revised. (Line 2-5)

Point 2: The abstract.

Response 2: It is really true as Reviewer suggested that we have revised abstract. (Line 15-27)

Point 3: Line 44-52

Response 3: We have re-written this part according to the Reviewer’s suggestion. (Line 34-66)

Point 3: Line 214

Response 2: We are very sorry for our incorrect writing and have revised. (Line 213)

Point 3: Line 279-280

Response 3: We have re-written this part according to the Reviewer’s suggestion. (Line 270-285)

Once again, thank you very much for your comments and suggestions.

Reviewer 2 Report

Date: September 9, 2020

Journal: International Journal of Molecular Sciences
Manuscript ID: ijms-914162-v2
Type of manuscript: Article
Title: Phylogenetic Relationship and Gene Rearrangement of the Complete Mitochondrial Genome of Cynoglossus monopus (Pleuronectiformes: Cynoglossidae), with a Comparative Analysis of Other Flatfishes
Authors: Chen Wang, Hao Chen, Silin Tian, Cheng Yang, Xiao Chen *
Submitted to section: Molecular Genetics and Genomics

Reviewer's report

The authors have sequenced complete mitochondrial genome sequence of the tongue sole Cynoglossus monopus. They found an interesting rearrangement of the genome studied. Combined with mitochondrial genomes from Genbank, they used the Cynoglossus monopus genome to reconstruct phylogenetic relationships of Cynoglossidae and Soleidae species.

The text and analysis were significantly improved. On the whole, I think the paper could merit publication in IJMS, but after minor revision.

Line 182: correct the text.

Figure 6 (legend, lines 202-203): change “… the third codon…” to “… the third codon positions…”.

Line 245: change “… the third codon…” to “… the third codon positions…”.

Line 262: change “…first and second codons…” to “…first and second codon positions…”.

Line 268: change “…a Brith-Death process …” to “… a Birth-Death process …”.

Line 284: consider revision: “…an instinct group…”.

Author Response

Dear Reviewers:

Thank you for your comments concerning our manuscript. Those comments are all valuable and very helpful for revising and improving our paper, as well as the important guiding significance to our researches. We have made correction which we hope meet with approval. Revised portion are marked in red in the paper. The main corrections in the paper and the responds to the reviewer’s comments are as flowing:

Point 1: Line 182: correct the text.

Response 1: We are very sorry for our negligence of the text and have revised. (Line 182)

Point 2: Figure 6 (legend, lines 202-203): change “… the third codon…” to “… the third codon positions…”.

Response 2: We are very sorry for our incorrect writing and have revised. (Line 202-203)

Point 3: Line 245: change “… the third codon…” to “… the third codon positions…”.

Response 3: We are very sorry for our incorrect writing and have revised. (Line 245)

Point 4: Line 262: change “…first and second codons…” to “…first and second codon positions…”.

Response 4: We are very sorry for our incorrect writing and have revised. (Line 262)

Point 5: Line 268: change “…a Brith-Death process …” to “… a Birth-Death process …”.

Response 5: We are very sorry for our incorrect writing and have revised. (Line 268)

Point 6: Line 284: consider revision: “…an instinct group…”.

Response 6: Considering the Reviewer’s suggestion, we have revised. (Line 284)

Special thanks to you for your good comments.